# Mechanical Flexibility of DNA: A Quintessential Tool for DNA Nanotechnology

**DOI:** 10.3390/s20247019

**Published:** 2020-12-08

**Authors:** Runjhun Saran, Yong Wang, Isaac T. S. Li

**Affiliations:** 1Department of Chemistry, Biochemistry and Molecular Biology, Irving K. Barber Faculty of Science, The University of British Columbia, Kelowna, BC V1V1V7, Canada; rs.narayan@ubc.ca; 2Department of Physics, Materials Science and Engineering Program, Cell and Molecular Biology Program, University of Arkansas, Fayetteville, AR 72701, USA; yongwang@uark.edu

**Keywords:** DNA bending, DNA stiffness, biosensor, DNA nanostructures

## Abstract

The mechanical properties of DNA have enabled it to be a structural and sensory element in many nanotechnology applications. While specific base-pairing interactions and secondary structure formation have been the most widely utilized mechanism in designing DNA nanodevices and biosensors, the intrinsic mechanical rigidity and flexibility are often overlooked. In this article, we will discuss the biochemical and biophysical origin of double-stranded DNA rigidity and how environmental and intrinsic factors such as salt, temperature, sequence, and small molecules influence it. We will then take a critical look at three areas of applications of DNA bending rigidity. First, we will discuss how DNA’s bending rigidity has been utilized to create molecular springs that regulate the activities of biomolecules and cellular processes. Second, we will discuss how the nanomechanical response induced by DNA rigidity has been used to create conformational changes as sensors for molecular force, pH, metal ions, small molecules, and protein interactions. Lastly, we will discuss how DNA’s rigidity enabled its application in creating DNA-based nanostructures from DNA origami to nanomachines.

## 1. Introduction

In the past two decades, DNA nanotechnology has put forth various remarkable structures and functions of DNA far beyond its role as the genetic material in living organisms. Owing to its outstanding properties (self-assembly, programmability, stability, resilience, etc.) DNA has been increasingly maneuvered in multiple different ways to propel the field of nanotechnology. DNA serves in multiple capacities, i.e., as aptamers, DNAzymes, molecular beacons, biosensors, nanoparticles, molecular machines, and nano-electronic components (e.g., nanowires, constituents of logic gates, etc.) [1]. One of the most fundamental features of DNA that has made many of its applications possible is its excellent mechanical rigidity against bending. Since the mechanical rigidity of DNA is sensitive to multiple intrinsic and extrinsic factors, it can be manipulated and employed for sensing applications.

A wide range of techniques has been used to determine dsDNA stiffness, persistence length (L_p_), curvature, and geometry. Peters et al., have comprehensively reviewed many techniques used for probing DNA curvature and flexibility in vitro as well as in vivo [2]. Some of the popular techniques include computational simulations [3,4,5,6,7,8,9,10], electrophoretic mobility assays [11,12,13,14,15,16,17,18], cyclization analysis [4,19,20,21,22,23,24,25,26,27,28,29,30,31], tethered particle motion assays [32,33,34], optical tweezers [35,36,37,38,39], fluorescence spectroscopy [40,41,42,43,44,45,46,47], atomic force microscopy [48,49,50,51,52,53], single-molecule force spectroscopy [54,55], cryo-electron microscopy [56,57,58,59], scanning Tunneling microscopy [60,61,62], X-ray crystallography [63,64], small-angle X-ray scattering [65,66], NMR spectroscopy [67,68,69,70,71,72,73], transient electric birefringence [74,75], anti-Brownian electrophoretic trap [76], chromosomal conformation capture (3C) assay [77,78,79,80], genetic and recombination assays [81,82,83,84,85,86,87,88,89,90,91,92,93], etc. Generally, a combination of two or more of the above methods is needed to gain a complete understanding of conformation and kinetics. To utilize the mechanical rigidity of DNA to our advantage, we must first understand the factors that influence it. Here, we are primarily interested in double-stranded DNA (dsDNA) due to its relatively high stiffness compared to single-stranded DNA (ssDNA). The persistence length (L_p_) of dsDNA is ~50 nm (at 10 mM Na^+^), over 60 times greater than that of ssDNA (L_p_~0.75 nm at 150 mM Na^+^) [94].

First, the prime mechanism underlying the dsDNA stiffness is the stacking interactions of adjacent base pairs that provide its structural integrity [75]. In a recent report, Chen et al., used computational methods to quantitatively show that base-stacking contributes significantly to the local and global rigidity of dsDNA as compared to other chemical interactions such as backbone, ribose puckering, and base-pairing in dsDNA [3]. Figure 1a illustrates the geometry of base-stacking interaction among other chemical interactions. Bending of the DNA directly perturbs the base-stacking geometry along the dsDNA, hence unfavourable. Therefore, factors that directly influence the base-stacking of dsDNA will directly impact DNA stiffness.

Second, the electrostatic phosphate-phosphate repulsions and environmental ionic strength also have substantial influence on the dsDNA stiffness. It is well known that asymmetric neutralization of phosphate charges induce bending of the helix axis [11,95,96,97,98,99]. Figure 1b is a cartoon showing that DNA spontaneously bends upon the incorporation of neutral phosphate analogs on one of the helical faces. Apart from the bending rigidity, the stiffness of the phosphate backbone also gives rise to dsDNA’s outstanding torsional rigidity (overwinding behaviour of dsDNA) [100,101]. The salt-dependence of the stiffness of dsDNA is closely related to the screening of the phosphate backbone charges as well. In 2019, Guilbaud et al., showed that the L_p_ of dsDNA decreases significantly with increasing ionic strength in monovalent and divalent metal ions (Figure 1c) [102]. This decrease in persistence length could be attributed to the reduction in the energetic cost of bending due to screening of phosphate backbone charges by surrounding cations. However, the L_p_ seems to be independent of the size of metal ions when kept at the same ionic concentration [102].

Third, the rigidity of dsDNA is sequence-dependent. In 2017, Jonathan et al., used computational simulations of random and λ-phage dsDNA fragments to report that the apparent L_p_ shows a standard deviation of 4 nm over the sequence. Additionally, they demonstrated that poly(A), poly(TA), and phased A-tract sequence motifs are exceptionally straight and stiff, tightly coiled and exceptionally soft, and exceptionally bent and stiff, respectively [103]. Figure 1d shows a scatter plot of simulated (Monte Carlo) and experimentally (includes data from atomic force microscopy, electron microscopy, and cyclization experiments) estimated poly(NN) dsDNA persistence lengths (L_p_) from different experiments from various studies. It is worth discussing here that nucleotide base modifications affect dsDNA stiffness. For example, 5-formylcytosine, 5-hydroxymethylcystosine and 5-hydroxymethyluracil are known to enhance the dsDNA flexibility while 5-methylcytosine is known to decrease it [54,104,105]. Figure 1e shows the fraction of the looped molecules as a function of time for unmodified dsDNA as well as dsDNA containing four copies of 5-formylcytosine (5-fC), 5-hydroxy-methylcystosine (5-hmC), 5-carboxylcystosine (5-caC), and 5-methylcytosine (5-mC). Higher flexibility of dsDNA leverages higher looping probability and faster looping kinetics. Figure 1e, shows the schematic of a single-molecule dsDNA looping experiment.

Fourth, dsDNA stiffness is strongly influenced by temperature. Through temperature-controlled single-particle tethered motion experiments, it has been shown that dsDNA stiffness is sharply reduced when temperature is increased (Figure 1f) [32]. The melting temperature of all three dsDNA used in Figure 1g are greater than 75 °C, while the experiment range is between 23–52 °C. This indicates that during the experiment the global conformation of all three DNA remains double-stranded. The authors have attributed this to local effects such as temperature-enhanced formation of kinks (due to unstacking of adjascent base pairs) and small melting bubbles (due to disruption of base pairing and/or base stacking). In line with the above discussion, it is worth emphasizing that basepair mismatches or melting bubbles significantly effect the rigidity of dsDNA [26,106,107,108]. In 2004, Yan et al., suggested that formation of melting bubbles by internal strand-sepraration provide a flexible-hinge that facilitates the formation of smooth bends and thus loops of dsDNA less than 150 bp [26]. In 2006, Yuan et al., demonstrated that distributed melting bubbles induce bigger bending angles and higher reduction in the stiffness of dsDNA as compared to the centrally located melting bubbles of comparable overall size [108]. In 2009, Forties et al., published a robust model that predicts the impact of small bubbles formed due to dsDNA melting (temperature-dependent) or DNA mismatches (sequence-dependent) on dsDNA flexibility [107].

Lastly, DNA-binding molecules such as bis-intercalators (e.g., YOYO-1) are known to alter dsDNA rigidity. Figure 1g demonstrates that as the concentration of YOYO-1 increases (in comparison to base-pair concentration), the fractional extension L/L_o_ of dsDNA increases (L and L_o_ are the contour lengths of dsDNA with and without YOYO-1), while the L_p_ decreases [48].

## 2. DNA-Springs as Regulators

### 2.1. Allosteric Regulation of Enzyme Activity

Regulation of protein or nucleic acid enzyme activity by factors such as pH, salt, temperature, and co-factor molecules have been widely investigated and understood. However, the effects of mechanical perturbation on enzyme activity remain under-explored. In the past decade, DNA-based molecular springs have been used to regulate the mechanical compliance of biomolecules by exerting mechanical strain to alter their conformation [109,110,111,112,113,114,115]. DNA spring is a bent dsDNA that stretches any molecule that bridges its ends, like a bow under tension. The concept of DNA springs was first introduced by Tyagi and Kramer in 1996 [116] (Figure 2a) where the conformational change in an DNA hairpin A (blue/yellow) was induced by a complementary DNA strand B (red). Here, hybridization of the loop of hairpin A (yellow) and strand B (red) into duplex A/B opens up the stem of hairpin A. As the stiffness of the double-stranded loop of hairpin A is stronger than that of the single-stranded loop of hairpin A, it acts as a spring and exerts force on the stem of hairpin A unzipping the double-stranded stem apart into single-stranded components. A more detailed discussion of the unzipping of hairpin stem due to formation of double-stranded DNA springs in hairpin loop is made in Section 3.1. Detection of this conformational change was achieved using fluorescence quenching using a fluorophore-quencher pair. The total fluorescence increases upon the formation of A/B duplex as the rigid dsDNA duplex keeps the fluorophore-quencher pair apart. DNA springs have since been employed in three different ways to probe the mechanical compliance of enzymes: (i) by influencing the spatial accessibility of molecules that regulate the enzyme activity, and by applying force directly on (ii) the enzyme or (iii) the substrate.

(i)*Influencing the accessibility of regulatory molecules:* The first strategy using DNA springs to regulate enzymes is by controlling the accessibility of enzyme regulatory factors, e.g., inhibitors. Ghadiri et al., reported the allosteric regulation of a protein enzyme Cereus Neutral Protease 5 (CNP) by a DNA spring using this strategy (Figure 2b) [117]. In this study, the two ends of DNA strand C (blue) were conjugated to CNP (shown as E) and its inhibitor (shown as I). The flexible nature of ssDNA C allows the inhibitor I to remain bound to the enzyme E until the complementary ssDNA D (red) hybridizes to strand C. The rigid dsDNA duplex C/D causes the dissociation of the inhibitor from the enzyme and keeps it away. CNP’s catalytic activity recovers noticeably upon the unbinding of the inhibitor I caused by the binding of strand B.(ii)*Mechanical force regulation of enzymes:* The second strategy uses DNA springs to exert force on enzymes to modulate their mechanochemistry. Choi et al., demonstrated this concept in 2005 (Figure 2c) [115] by covalently attaching two ends of a flexible 60-nucleotide ssDNA strand *F* (blue) to the two lobes of a maltose-binding protein (MBP). MBP retains its high binding affinity to maltose when *F* remains in the ssDNA state, as it does not apply external force to the MBP. However, when a complementary strand *G* binds to *F*, an external force up to 10 pN is applied on the MBP from the stiffness of the *F/G* duplex. This mechanical stress makes it energetically unfavourable for MBP to undergo the conformational changes required for maltose binding. As the length of strand *G* increases above 30 bases, the mechanical stress exerted on MBP increases, leading to further diminished maltose-binding affinity. Through this mechanism, regulation of MBP-maltose affinity was achieved through the external forces generated by a DNA spring. A similar concept was applied to achieve reversible allosteric regulation of enzyme guanylate kinase (GK) [109] and cAMP-dependent Protein Kinase A (PKA) [110]. The binding of GK to its substrates adenosine triphosphate (ATP) and guanosine monophosphate (GMP) requires a conformational change, which was reversibly inhibited by mechanical tension applied through a dsDNA spring. On the other hand, PKA was activated by the mechanical tension applied through a dsDNA spring, such that it demonstrated activity even in the absence of cyclic adenosine monophosphate (cAMP), the co-factor typically required for its activation. In 2007, Silverman et al., extended the application of DNA springs to the allosteric regulation of ribozymes (Figure 2d) [118]. Here, the hammerhead ribozyme’s mechanical control was attained by attaching two complementary ssDNA *G* and *H* to two different portions of the ribozyme. When *G* and *H* hybridize, they pull apart their corresponding attachment points, causing the tertiary structure of ribozyme *E* to unfold and lose function.(iii)*Mechanical force regulation of substrates:* The third strategy is to exert strain on the substrate such that it affects the efficiency of the enzyme. Based on this strategy, a circular ssDNA was used as a topological constraint to regulate the Rolling Circle Amplification (RCA) activity of Φ29 DNA Polymerase enzyme [119]. This was illustrated by Liu et al., where a circular RCA ssDNA template (shown as *S*, blue) was mechanically strained by a strong linking duplex formed with another circular ssDNA (shown as *J*, red) as in the case of DNA catenanes (Figure 2e). The strained template cannot undergo RCA until the spring/constraint is cleaved, and the strain is released. This system has been employed to demonstrate specific target-triggered RCA for detecting a specific *E. coli* strain with detection limits of 10 cells/mL.

### 2.2. Regulation of Live Cells

DNA-springs can be made to regulate the activity of not only individual biomolecules but also live cells. In 2017, Zhang et al., demonstrated the use of a DNA nano-spring for the reversible control of integrin clustering and cell membrane receptor function (Figure 2f) [120]. Here, a DNA nanospring was demonstrated to redirect the normal morphology of the cell to having multiple cell protrusions and even alter the mRNA expression levels of integrin-related genes. In this study, DNA nano-springs consist of long repeats of hairpin-forming sequences (generated by RCA) were used as a scaffold for assembling RGD-DNA sequences (shown in orange). RGD is a tripeptide (Arg-Gly-Asp) with a high affinity for cell adhesion through integrin. Upon the addition of external DNA sequences (shown as DNA1 and DNA2), the hairpins can undergo cycles of unhybridization/rehybridization. This stretches and contracts the distance between RGD and thus the distance between and force through the integrins. When the nanospring is contracted, the clustered integrins stimulate the cells to form focal adhesions. While in the extended state, increased distance between integrins triggers activation of PI3K/Rac1 signalling, causing membrane remodelling and generation of numerous cell protrusions. Regulation of cell adhesion has also been achieved through the rational design of DNA tethers with various mechanical stability levels. In these studies, complementary strands of dsDNA were conjugated to the surface and cell-adhesion receptors (e.g., RGD or Selectin), where they act like nano-springs with a tension tolerance. When external forces increased beyond a certain threshold, the DNA springs would break. This system has resulted in a series of studies leading to quantifying integrin adhesion force, notch receptor activation, and leukocyte rolling adhesions [121,122,123,124,125,126,127]. Lastly, regulation of subcellular membrane remodelling was achieved with the help of DNA origami nano-springs as reported in 2019 by Grome et al. [128]. Here, DNA nano-springs of varying structural stiffness were polymerized on the surface of the liposomes and were found to significantly influence membrane binding, membrane remodelling, as well as vesicle tubulation.

## 3. Mechanical Rigidity-Facilitated DNA Sensors

### 3.1. DNA Springs as Force-Sensors

Using DNA as springs for sensing nanoscale forces was introduced by Shroff et al., in 2005. Herein a FRET-based force sensor capable of sensing a force range of 0–20 pN was demonstrated using a circular ssDNA *B* (red) covalently attached to a FRET donor D (green) and acceptor A (yellow) (Figure 3a) [129]. Change in FRET occurs when the circular ssDNA *B* converts to dsDNA through hybridization with a complementary ssDNA *C* (blue) of different lengths (Figure 3a). The change in FRET measures the force applied through the DNA-spring on the ssDNA. The idea of hybridization-induced stress and strain on circular DNA was further used to assess the force sensitivity and nanomechanics of several systems. In 2013, Fields et al., illustrated a similar concept to control strain in a DNA hairpin system (Figure 3b) [130]. A DNA-vise was constructed from a hairpin *B* with a loop of 30–50 nucleotides and a stem of 49 base pairs. Hairpin *B* was tagged with a FRET-pair at its loop-stem junction, such that the unzipping of the stem can be monitored by the decrease of FRET. As more nucleotides within the hairpin loop hybridize with longer complementary ssDNA *C (l)*, the stress increases to unzip the hairpin from the loop side. Within a range of *l* values (*l* < *l_buckle_*), increasing stiffness of *B/C* duplex results in increasing unzipping of the stem of hairpin B as monitored by the decrease of FRET. However, at lengths above an Euler-buckling threshold (*l > l_buckle_*) the base-pairing free energy to re-establish the full hairpin stem overcomes the bending energy of the DNA-spring, causing FRET to return to a high state.

Lately, the increased complexity of 3D DNA origami is being used for constructing origami-based nano-springs to leverage extremely sensitive DNA-based force-sensors. In 2013, Zhou et al., fabricated a DNA origami compliant joint structure, which acts as a tunable mechanical nano-spring [131]. The balance of tension in the flexible ssDNA components at the joint plays a key role in determining its geometry and mechanical properties. This study demonstrated the possibility of creating more elaborate DNA-origami sensors based on the mechanical flexibility of DNA. A DNA origami-based force-spectrometer was reported in 2016 by Funke et al. [132] where the fluctuations of the DNA spring-loaded hinge sensor allowed the measurement of inter-nucleosome distance at sub-nanometer resolution. Figure 3c shows the schematic of the DNA origami-based force spectrometer, which comprises of a spring-loaded hinge and two nucleosomes attached on opposite sides of the hinge. The torque generated by the hinge is shown by a red torsional spring. A FRET-pair was attached to the opposite sides of the spring-loaded hinge to gauge the distance between the nucleosomes. It is known that nucleosomes condense into arrays, which indicates that attractive forces exist between individual nucleosomes. The DNA origami-based force spectrometer, shown in Figure 3c, was used to measure the interaction strength between two nucleosomes directly. In this system, the strength of nucleosome-nucleosome interactions resulted in inter-nucleosome distance changes. These changes in distances were reflected by changes in FRET between the acceptor and donor fluorophores. In 2016, Iwaki et al., developed a programmable DNA origami-based nanospring that enabled the monitoring of force-induced transitions between two structurally distinct states of the mechanosensory motor protein myosin VI with nanometer-precision [133]. The DNA nanospring in this study comprises a 7,308 nucleotide ssDNA and more than 150 species of short ssDNA (<50 nucleotides) self-assembled into a two-helix bundle that is strained with negative superhelicity to form a coil structure. The DNA nanospring was attached to immobilized myosin II on one end and myosin VI on the other end. Figure 3d shows myosinVI moving unidirectionally along an actin filament against the load of the nanospring. This system helped study the stepping dynamics of Myosin VI under tension using fluorescently labelled DNA nanospring and TIRF microscopy. Lastly, DNA spring has been used in mechanical regulation and monitoring of biomolecules in live cells, taking advantage of the difference in rigidity of dsDNA vs ssDNA, for developing tools to study cell mechanics [134,135,136,137].

### 3.2. pH-Sensitive DNA Spring

In 2010, Wang et al., constructed a G-quadruplex/i-motif-based DNA nanospring powered by protons, which exhibited highly sensitive pH-dependent spring-like structural changes (Figure 4) [138]. The nanospring consists of two building blocks called subunit I and II, each of which is made up of two circular ssDNA C connected via ssDNA A and B. The subunits I and II can be connected by strand D in a linear array. The ssDNA A, B, and D consist of four G-rich stretches and form G-quadruplexes. The formation of G-quadruplexes reduces the distance between the circular DNA C rendering the nanospring in a compressed state. Upon adding a C-rich strand E, the strands A, B and D hybridize to strand E. Formation of the duplexes A/E, B/E, and D/E increases the distance between circular DNA C, imposing an extended state of the nanospring. This is due to the fact that in the single-stranded form the strands A, B and D have low rigidity and are floppy, while in the double-stranded form the A/E, B/E, and D/E are rigid due to their relatively increased persistence length. In the presence of slightly acidic conditions (~pH 6.0 or less), the cytosines are partially protonated, and the C-rich strand E un-hybridizes from strands A, B and D and folds into a closed i-motif structure, freeing the G-rich ssDNA strands A, B and D to form G-quadruplexes again. When the pH rises back to the alkaline conditions (~pH 8.0) the cytosines get deprotonated, leading to destabilization of the i-motif. This leaves the strand E free to hybridize back with strands A, B and D, which brings the nanospring back into the extended state. The above-explained DNA nanospring is pH-responsive, and the extended or compressed state can indicate a change between acidic and alkaline pH.

### 3.3. DNA-Bows as Metal Ion/Small-Molecule Sensors

The interactions of DNA with metal ions and small molecules facilitate fundamental cellular processes [139,140,141,142] such as genomic stability [139,143,144,145,146,147], DNA-carcinogen interactions [146,147,148,149,150,151], drug development [152,153,154,155], and DNA-based metal-biosensors [156]. As mentioned earlier, the rigidity of DNA is affected by the type and concentration of metal ions. This mechanical energy stored in bent-DNA molecules has been utilized by Freeland et al., to quantify the interaction between DNA and metal ions (M^n+^) and small organic molecules (Mols) using gel electrophoresis [17,18]. Their cost-effective and straightforward strategy utilizes the mechanical energy stored in bent DNA molecules called DNA-bows (Figure 5a) [17,157,158,159]. Each bow consists of two DNA strands with one part as a bent double-stranded fragment (F1) and the other as a mechanically stretched single-stranded fragment (F2) (Figure 5a). Depending on the environmental conditions, DNA-bows may decrease their bending elastic energy by transitioning to relaxed DNA-dimers or relaxed DNA-multimers (R), which show different electrophoretic mobilities (Figure 5b,c) for quantification. Freeland et al., reported that the equilibrium between the ss-DNA (S), DNA-bows (B) and multimers (R) is perturbed by the presence of metal ions and small organic molecules in a concentration-dependent manner, allowing for the quantitative determination of their interaction strength with DNA (Figure 5). This was indeed measured for various metal salts (MgCl_2_, MgSO_4,_ KCl, CaCl_2,_ Al(NO_3_)_3,_ Zn(NO_3_)_2_, and AgNO_3_) and small organic molecules (guanidine, putrescine, spermidine, ethidium bromide, SYBR safe, ganciclovir, and thiamine). (Figure 5d,e) Undoubtedly, DNA-bows demonstrated the potential for developing sensitive and economical metal ions sensors to screen metal ion-aptamers, DNA-targeting drugs, and DNA-protein interactions in general.

### 3.4. DNA Stiffness Assisted Temperature and Osmolarity Sensing

The sensitivity of DNA towards changes in temperature and osmolarity are well known. In prokaryotes, the Histone-like nucleoid structuring protein (H-NS) is an abundant protein that plays a vital role in regulating nucleoid structure [160,161,162], in gene expression [163,164,165,166,167,168,169], and in mediating cellular response to changes in metabolite pH, temperature, and osmolarity [161,170,171]. It exhibits preferential binding to A & T-rich sequences and other regions of high intrinsic curvature, along the backbone of double-stranded DNA [162,166,172]. H-NS consists of two distinct domains (C-terminal and N-terminal) connected by a flexible linker segment. While the C-terminal domain bears a unique DNA binding motif, the N-terminal domain consists of a coiled-coil motif that mediates H-NS oligomerization [72,172,173]. Multiple studies suggest that H-NS bends dsDNA upon binding [174] and strongly confer dsDNA compaction [175,176]. However, it has also been shown that at high concentrations, H-NS binds to DNA from end-to-end with no significant compaction [162]. In 2003, using single-molecule force spectroscopy [177,178], Stavans et al., reported that at physiological concentrations, H-NS binding to λ-DNA covers extensive DNA-tracts, causing the contour length of the λ-DNA to extend 2-fold and DNA stiffness (measured by L_p_) to increase 3-fold (Figure 6a) [179]. The result indicated that each H-NS dimer occupies 15–20 base-pairs along the λ-DNA, which was further supported by a structural investigation of the H-NS/DNA complex by using X-ray crystallography [180].

Stavans et al., showed that H-NS’s polymerization on DNA is sensitive to changes in osmolarity and temperature. The *L*_p_ of λ-DNA increases with increasing H-NS concentration and decreasing ionic strength (Figure 6b), and decreases with temperature (Figure 6c) up to 32 °C. However, at 37 °C, H-NS no longer induces any rigidity change of DNA. Not only do the above results provide remarkable insights into the mechanism of N-HS mediated gene silencing in vivo, but the system also demonstrated great potential as an in vivo osmolarity and temperature sensor for subcellular local environment.

### 3.5. DNA Bending Assisted Protein Sensing

Histone-like proteins in prokaryotic cells and transcription factors (TFs) are essential architectural and functional regulators of genetic materials in cells [181,182]. Understanding of their function and regulation requires sensitive strategies to detect their interactions with DNA [183,184,185,186,187,188]. In 2004, Shen et al., reported an elegant DNA-based nanomechanical sensor that involves bending of DNA upon binding to *E. coli* Integration Host Factor (IHF), a histone-like protein [189]. The device consists of a DNA double-helical shaft (consisting of the IHF binding site) connecting with two rigid DNA triple crossover (TX) motifs (Figure 7a). The two TX motifs are individually labelled with a FRET donor D (green) and acceptor A (yellow) [190].

The FRET signal decreases as the IHF binds and distorts its binding site, increasing the distance between the FRET-pair (Figure 7a). Shen et al., have further extended this system by connecting the TX motifs with a pair of complementary DNA strands (Figure 7b). If the IHF binding free energy exceeds the DNA hybridization free energy, IHF binding is energetically favoured (Figure 7b). Therefore, when the length of complementary base-pairs increases, the ability of IHF binding to the structure decreases as the overall free energy favours the DNA-bound structure more than IHF-bound structure. With this method, the device was used to estimate the binding free energy between IHF and DNA.

DNA-bending is one of the most significant mechanisms employed in TF-mediated gene modulation [192,193]. In 2012, Crawford et al., developed a DNA bending-based TF biosensor that can discriminate between the TF’s active and inactive forms. As a proof-of-concept, their system employs a FRET-based detection of low concentrations of Catabolite gene Activator Protein (CAP). The dsDNA sensor designed by Crawford et al., consists of three 5-adenine (A_5_) kinks (5 unpaired adenines) around a CAP-binding site (Figure 7c). The unpaired adenines in the A_5_ kinks confer specific sequence-directed bends, i.e., DNA bend angle of 73° ± 11° [194] bringing the fluorophore pair at the ends of the sensor DNA within the FRET range (Figure 7c). As shown in Figure 7c, the addition of CAP moves the ends farther apart, consequently reducing FRET in a concentration-dependent manner and enabling the detection of low concentrations of CAP. This system can be extrapolated appreciably to many more TFs, which marks significant progress in TF sensing.

## 4. DNA-Flexibility: A Game-Changer for DNA Nanostructures

The field of DNA origami is progressing rapidly and has produced nanostructures and nanomachines with fascinating applications. The 3-dimensional geometry of DNA origami nanostructures largely depends on the structural rigidity of the constituent DNA molecules [195]. In order to predict the shape and rigidity of origami in solution, multiple state-of-the-art computational modelling methods have been developed that can compute origami features like DNA bending, stretching, twisting, stiffness, and elasticity [196,197,198,199]. We will focus on two of the many frontiers in DNA origami research, namely the “circular DNA-based origami” and “DNA nanomachines”. With the examples discussed below, we aim to highlight how they heavily take advantage of the mechanical rigidity of DNA. We also highlight how small perturbations in the mechanical rigidity of DNA brings a significant change in the efficiency of functional DNA nanostructures.

### 4.1. Circular DNA as the Basis of Origami

One of the most captivating categories of bent or curved DNA is that of circular DNA. Although commercial synthesis of circular DNA is expensive for day-to-day experiments, various protocols are available for circularizing linear DNA. The propensity of a DNA to circularize is directly related to its sequence- and length-dependent mechanical rigidity. Cyclization propensity (calculated as the Jacobson-Stockmayer *J* factor) is a well-established method for experimentally and computationally probing a DNA fragment’s rigidity [4,130,200,201]. DNA nanotechnology has exploited the flexibility-based cyclization of DNA fragments to construct fascinating DNA origami nanostructures. These include circular interlocks, single- and multiple-ring DNA catenanes, Borromean rings, and rotaxanes [202,203,204,205]. Circular functional nucleic acids have been employed as rolling circle amplification templates, aptamers, enzymes two-input logic gates, biosensors and have proved to be a cornerstone for the advancement of DNA nanotechnology [206].

### 4.2. Mechanical Flexibility of DNA Can Tune the Efficiency of DNA-Based Nanomachines

Recent advances in artificially synthesized molecular machines have massively taken advantage of the structural and mechanical properties of DNA. DNA has been employed as a critical component of various types of synthetic molecular machines and of the molecular tracks which the machines navigate [207,208,209,210]. Nevertheless, these synthetic machines have yet not matched the biological macromolecular machines in terms of their vital characteristics, such as directionality, step size, processivity, speed, and chemical yield. Since DNA nanomechanics plays a crucial role in the systems overall free energy, it is intuitive to think that tuning the flexibility of the constitutive DNA may have a significant impact on the performance of such machines.

Tomov et al., have demonstrated the significance of DNA flexibility in the efficiency of functional DNA nanostructures through experiments and molecular simulations of a DNA bipedal motor’s walking dynamics [211,212,213,214,215]. For example, the foothold and the walker legs in their current design are shorter than the *L*_p_ of ds DNA and need to be made more flexible to achieve a larger step size (Figure 8a). However, the increase in such flexibility also enhances the search-space volume available to the walker, thus increasing the activation barrier to leg-placing and reducing the stepping yields. Additionally, they also pointed out that the walker’s efficiency is sensitive to the mechanical properties and curvature of the DNA origami track. They further pointed out that their design of the 2D origami sheet bears a curvature as the origami sheet’s effective designed pitch (10.67 bp/turn) is larger than that of real DNA (10.5 bp/turn). They suggested reducing this origami bending or curvature into a stiffer walking track will assist the walker in moving in the intended direction (Figure 8b).

It is a known fact that while the average persistence length (L_p_) of dsDNA is ~50 nm and increases significantly with increasing GC content, the local persistence length (L_p_’) of dinucleotide steps ranges from 40–55 nm [216,217]. In a unique computational study, Park et al., have exploited the sequence-dependent flexibility of DNA to propel a DNA-based Brownian ratchet for directional transporting positively charged nanoparticles [212]. In this study, Brownian dynamics simulations of coarse-grained models have been implemented to construct a single, 130 nm long, dsDNA with its sequence-dependent local flexibility gradually increasing along its length at the physiological salt concentration (0.15 M) (Figure 8c). Further, a positively charged nanoparticle (NP) is simulated to bind to this DNA. The DNA-NP binding causes the DNA to bend (Figure 8d), which augments the effect of the variation in sequence-dependent DNA flexibility. This DNA flexibility gradient creates an asymmetric potential for the DNA-NP binding and fuels the directional and processive motion of the NP towards the higher flexibility region on the DNA. However, upon increasing the salt concentration to 0.81 M the DNA-NP binding does not induce significant bending (Figure 8e), and thus this binding is less dependent on DNA flexibility. In this case, there is a negligible gradient; therefore, the NP diffuses randomly in either direction on the dsDNA. The DNA-NP system designed in this study has this 130 nm long fragment repeated such that the asymmetric potential is repeated periodically in a single, long dsDNA molecule. By repeatedly switching the salt concentration between 0.15 M (C_on_) and 0.81 M (C_off_) over several cycles, the directional and processive motion of NP is demonstrated.

In one of the latest reports, Suzuki et al., have taken advantage of serially repeated tension-adjustable modules to induce large reversible structural deformations in a DNA-origami nano arm (fabricated as an eight-helix DNA bundle) [218]. The tension-adjustable modules are composed of a stem (4 helices), stiff piers (>4 helices including the stem), and bridge strands. The deformation of the nano arm into increasingly complex shapes has been demonstrated by placing varying amounts of tension in its flexible DNA modules.

## 5. Summary and Future Directions

In summary, dsDNA’s mechanical rigidity has been used in a wide range of nanotechnology applications, from understanding and controlling biologically active molecules to probing cellular mechanics and creating nanomachines. The free energy required to bend a piece of dsDNA adds a new dimension in controlling the conformational states of the molecule in addition to the traditional base-pairing interactions. However, there are still only a few examples that fully utilize the mechanical bending of dsDNA in sensing applications at the current stage. We foresee that incorporating DNA bending as a functional element to the existing designs can create new generations of DNA-based nanodevices and sensors with bifunctional properties and extended sensitivity range in the near future.

## Figures and Tables

**Figure 1 sensors-20-07019-f001:**
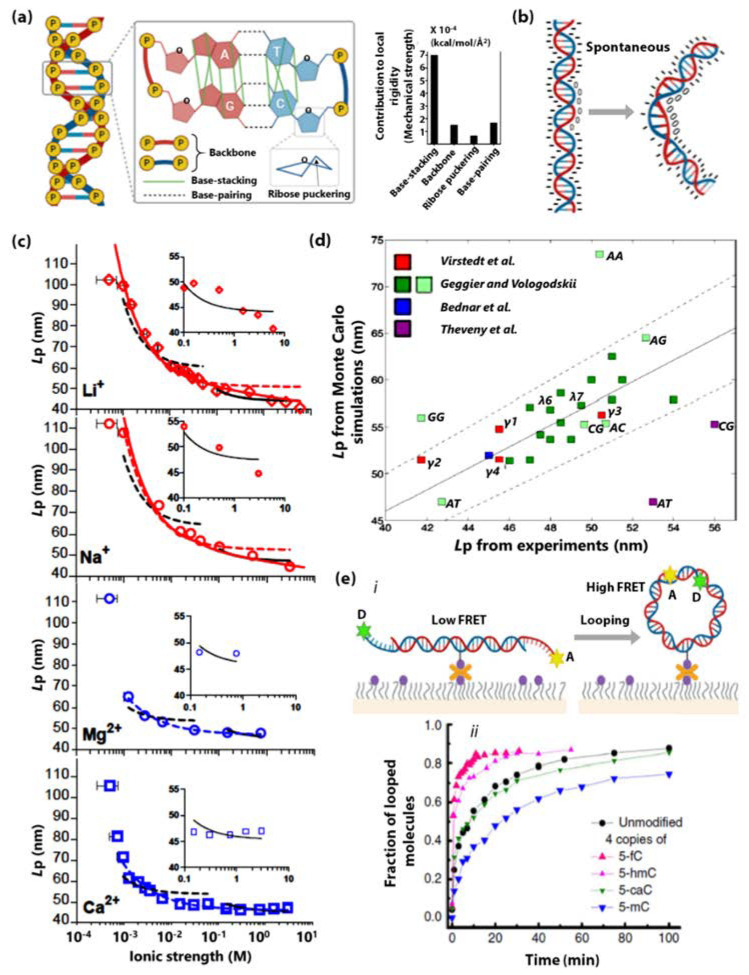
(**a**) A schematic of dsDNA showing chemical interactions and structural features such as base-pairing, base-stacking, phosphate-backbone, sugar pucker, respectively, and their contribution to the local rigidity in terms of mechanical strength of dsDNA. Figure adapted and modified from [3] with permission from The Royal Society of Chemistry. (**b**) Cartoon representation of dsDNA spontaneously bending upon incorporating neutral phosphate analogs (shown as ‘0’) on one of the helical faces. Negatively charged phosphates are shown as ‘-’ [11]. (**c**) Dependence of L_p_ on the ionic strength of monovalent Li^+^, Na^+^ (top red) and divalent Mg^2+^, Ca^2+^ (bottom blue) ions. Figure reprinted with permission from [102]. Copyright (2019) by the American Physical Society. (**d**) A scatter plot of simulated and experimentally estimated poly(NN) dsDNA persistence length (L_p_) data from different experiments pursued by various research groups (grouped by colour). Figure adapted with permission from [103]. Copyright 2016 American Chemical Society. (**e**,**i**) shows the schematic of a single-molecule dsDNA cyclization experiment. Two complementary sticky ends of the dsDNA are tagged with a fluorescence resonance energy transfer (FRET) pair of fluorescent donor (D) and acceptor (A) dyes, and the fraction of the dsDNA looped is probed by monitoring the increase in FRET between A and D. (**ii**) shows the fraction of the dsDNA looped as a function of time. Figure adapted with permission from [23]. Copyright 2016, the authors. (**f**) Dependence of L_p_ of three dsDNA (differing in their G-C content) on temperature. The melting temperature of all three dsDNA used in this experiment are greater than 75 °C. Figure adapted with permission from [32]. Copyright 2014 American Chemical Society. (**g**,**i**) shows the change in dsDNA length upon YOYO intercalation. L and L_o_ is the contour length of dsDNA in the presence and absence of YOYO, respectively. The curve demonstrates the increase in fractional extension L/L_o_ upon an increase in YOYO concentration. (**ii**) shows the dependence of L_p_ on fractional extension L/L_o_. The curve demonstrates that L_p_ decreases upon the increase of the YOYO/base-pair ratio. Figure adapted from [48] with permission from The Royal Society of Chemistry.

**Figure 2 sensors-20-07019-f002:**
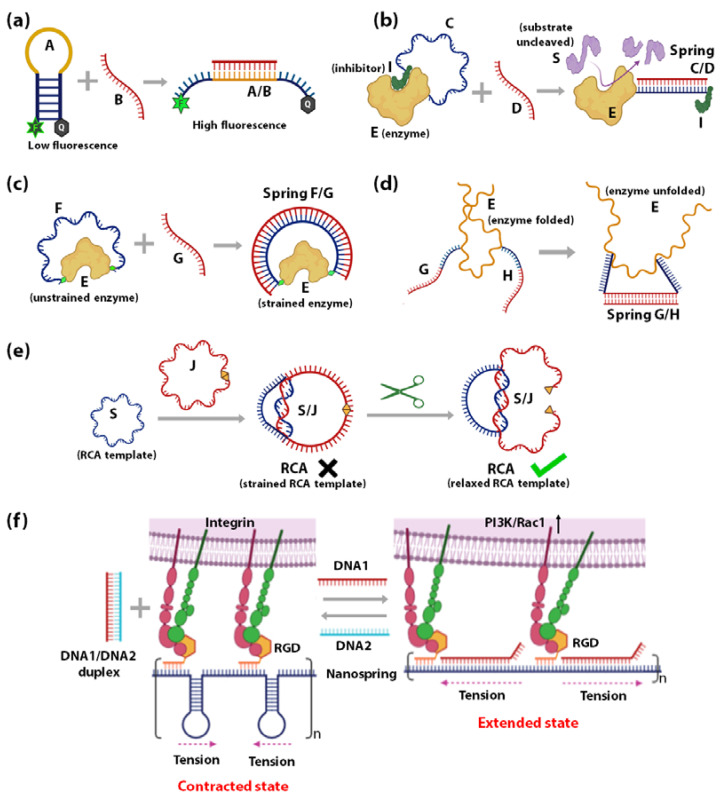
(**a**) Schematic representation of a DNA spring where the increased stiffness of A/B duplex does not allow the two ends of strand A to self-hybridize [116]. Schematic representation of a DNA spring regulating the catalytic activity of enzyme E (**b**) by altering the spatial proximity of the enzyme-inhibitor I with the help of the stiffness of C/D duplex [117] (**c**) by exerting mechanical force (generated due to the stiffness of F/G duplex) on the enzyme itself [115] (**d**) by pulling apart the ribozyme E tertiary structure causing it to misfold by the formation of G/H duplex [118]. (**e**) Schematic representation of a topological constraint J regulating the activity of an RCA DNA polymerase by exerting strain on the RCA template S by S/J duplex formation [119]. (**f**) Schematic representation of a DNA nanospring regulating mechanical tension by contracting membrane-bound integrins [120]**.**

**Figure 3 sensors-20-07019-f003:**
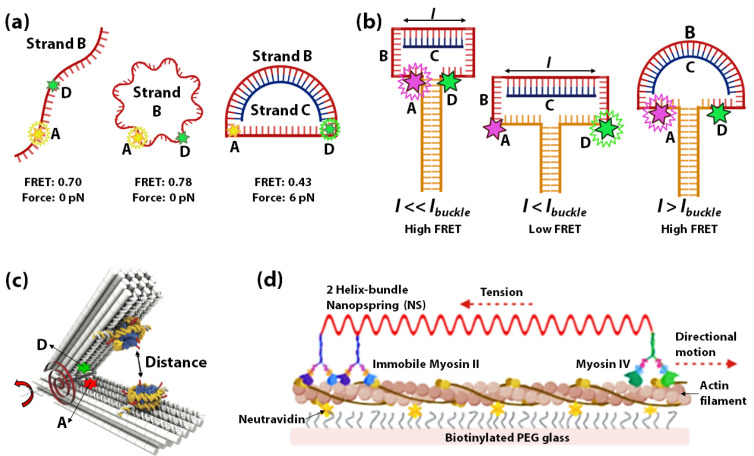
Schematic representation of force exerted by DNA spring (B/C duplex) on (**a**) the single-stranded part of circular ssDNA B [129] and (**b**) the stem of hairpin B [130]. Increasing force increases the distance between the fluorophores D and A, resulting in a decrease in FRET (by unzipping in case of (**b**)). (**c**) Schematic representation of the DNA origami-based force-spectrometer consists of a spring-loaded hinge bearing two attached nucleosomes and a FRET pair of acceptor (A) and donor (D) fluorophores. The red torsional spring depicts the torque generated by the hinge. The nanospring is used for sensing distance (and interactive forces) between nucleosomes. Figure adapted with permission from [132]. Copyright 2016, the authors. (**d**) Schematic representation of a two-helix bundle DNA nanospring tethered to immobilized myosin II on one end and myosin VI on the other. Myosin VI is shown to more unidirectionally along actin filament against the load of the DNA nanospring [133].

**Figure 4 sensors-20-07019-f004:**
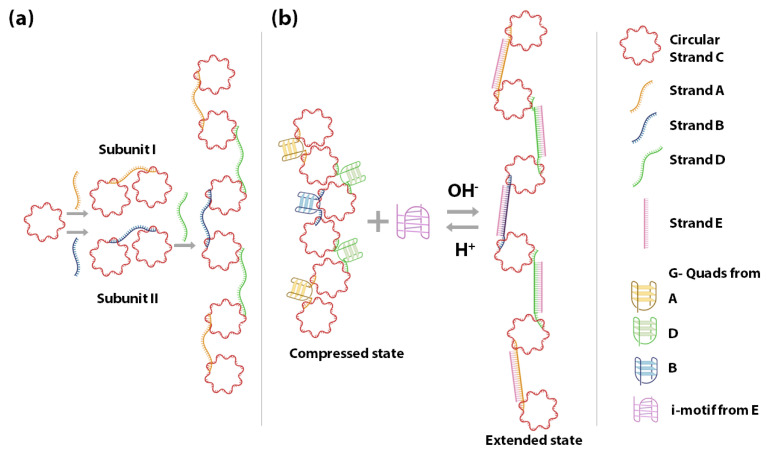
Schematic representation of the proton-fueled DNA nanospring. (**a**) shows the constituent DNA strands and the construction of the nanospring. (**b**) shows the operation of the nanospring with the help of the C-rich strand E, and its compressed and extended state in response to acidic and alkaline pH, respectively [138].

**Figure 5 sensors-20-07019-f005:**
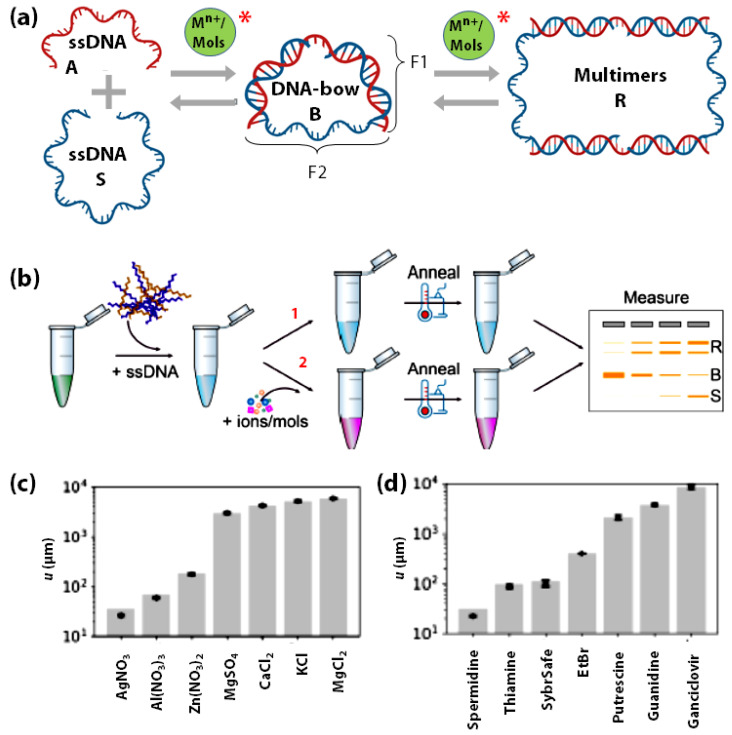
Schematic of (**a**) the constitution of DNA-bows from single-stranded DNA, and the perturbation (depicted by *) of the equilibrium between ss-DNA, DNA-bows, and multimers due to their interaction with metal ions (M^n+^) or small organic molecules (Mols), (**b**) the procedure followed for sensing DNA-metal ion/small molecule interaction via gel electrophoresis, where ‘1’ indicates the negative control with no Mn+/’Mols’ added while ‘2’ depicts the actual experiment with different concentrations of M^n+^/’Mols’ added. Fitted u-values for quantifying the strength of DNA interactions with (**c**) metal ion salts and (**d**) small organic molecules. (**b**,**c**) are adapted with permission from [18]. Copyright 2020, by the authors.

**Figure 6 sensors-20-07019-f006:**
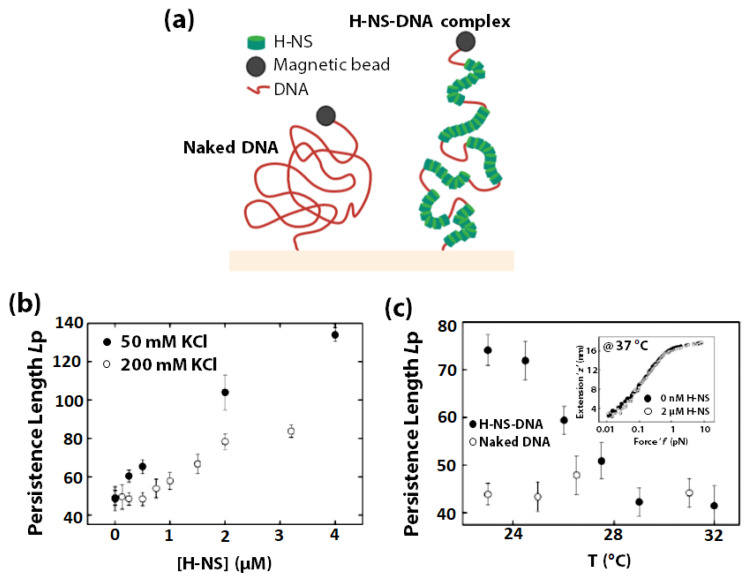
The effect of osmolarity and temperature on H-NS/DNA complex was investigated through force-extension experiments. (**a**) Schematic representation of the extended configuration of H-NS-λ-DNA complexes compares to that of naked λ-DNA in the low-tension regime (≤10 pN). The setup for measuring the end-to-end extension of DNA as a function of stretching force (magnetic force) is shown where DNA is tethered to a magnetic bead on end and a glass surface on the other. Change in persistence length (L_p_) of (**b**) H-NS-DNA as a function of increasing H-NS concentration in the presence of 50 mM (solid circles) and 200 mM (open circles) KCl, (**c**) H-NS-DNA complexes (solid circles) and naked DNA (open circles) with increasing temperature. Inset: Extension ‘z’ in nm (y-axis) of single H-NS-DNA complexes as a function of force (f) in pN (x-axis) measured at 37 °C for 0 (solid circles) and 2 µM (open circles) H-NS. (**b**,**c**) adapted with permission from [179]. Copyright 2003, Biophysical Society.

**Figure 7 sensors-20-07019-f007:**
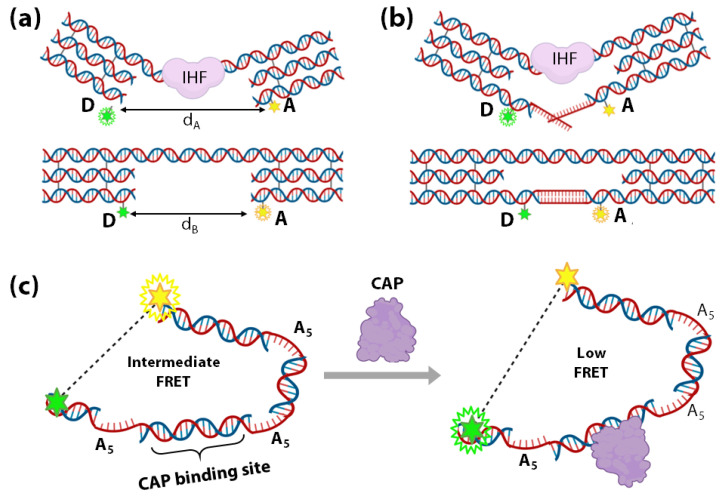
(**a**,**b**) Schematic representation of a nanomechanical device with the double helices as rectangular boxes and the TX motifs as three infused rectangular boxes. The upper domain connects the two TX motifs with the binding site for IHF. Each TX motif is labelled with either FRET donor fluorescein (D) or FRET acceptor Cy3 (A). IHF (purple) binding distorts the connecting shaft in the lower panel, consequently increasing the distance between D and A (d_B_ to d_A_) and thus decreasing FRET. (**b**) shows that the bottom domains are extended and connected by a cohesive tract [189]. DNA bending-based TF (CAP) sensor. (**c**) Left: Schematic of the sensor constituted by a dsDNA consisting of three A_5_ kinks and a CAP binding site (indicated by the blue dotted line). The donor (glowing green star) and the acceptor (glowing yellow star) fluorophores are depicted to be attached on either end of the sensor dsDNA, respectively. Right: Schematic of the pulling far apart of both the fluorophore-tagged ends of the dsDNA sensor from each other due to DNA bending induced by CAP binding [191].

**Figure 8 sensors-20-07019-f008:**
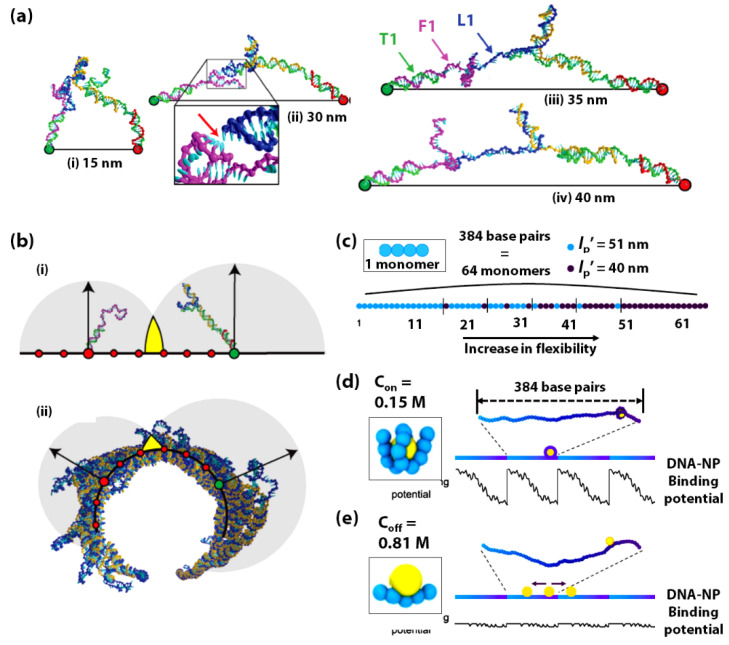
(**a**) Brownian dynamics simulations of coarse-grained models of the transition state configurations of the DNA bipedal walker, illustrating the increasingly stretched nature of the walker at (i) 15 nm, (ii) 30 nm, (iii) 35 nm, and (iv) 40 nm step sizes, with T1, F1, and L1 being one of the two the ‘foothold strand’, ‘Fuel strand’, and ‘Leg strand’. The red arrow in (ii) indicates the position of a leg-placing base pair. (**b**) Schematic of the DNA bipedal walker (blue-orange) with (i) one foot up and the other down on a flat surface, and (ii) an axial view of the configuration of the 2D origami sheet curled up into a tubular structure, with the red circles denoting the short axis foothold positions. The step size (~32 nm) is denoted by the larger green and red circles, which is the same for both (i) and (ii). The gray circles in both (i) and (ii) represent the regions of space well-sampled (95% of configurations have their terminal base within the sphere) by the unbound section of the walker (blue) and the foothold-bound fuel (purple) when the secondary structure was forbidden in the simulations. Arrows represent the radii of the spheres, and the yellow colour indicates the overlap between the spheres. (**a**,**b**) are adapted with permission from [211]. Copyright 2017, the authors. (**c**) Schematic of the dsDNA fragment modelled by a chain of 64 monomers (1 monomer = 6 basepairs) with flexibility gradient. The local persistence length (*L*_p_’) is set for each monomer at either 51 or 40 nm. The sequence-dependent dsDNA flexibility is gradually increased by increasing the fraction of monomers with *L*_p_’ = 40 nm. Schematic of the long dsDNA containing four repeat fragments with the flexibility gradient, together with its DNA-NP binding potential energy along the entire dsDNA molecule at solution salt concentrations of (**d**) Con = 0.15 M and (**e**) Coff = 0.81 M, respectively. The inset in (**d**,**e**) shows the specific representative configuration of DNA-NP complex for which the bending energy is calculated. (**c**–**e**) are adapted with permission from [212]. Copyright 2019, the authors.

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
