# Peer review of "Mechanical Flexibility of DNA: A Quintessential Tool for DNA Nanotechnology"

_sensors, 2020, doi:10.3390/s20247019_

Round 1

Reviewer 1 Report

In this review article, Saran, Wang, and Li highlight recent examples of using DNA nanoswitches to mechanically control the activities or assembly of other biomolecules. These devices utilize the change in stiffness between single-stranded and double-stranded DNA. I appreciate the authors’ effort in compiling a diverse array of applications and presenting them under one unifying theme and would like to see this article published. The article can be improved in the order of presentation, wording, referencing, and the quality of figures (fonts). Specific comments are listed below.   

13: structures conformation -> structure formation (perhaps?) 

42: The sentence reads like reference 4 revealed for the first time that base stacking contributes significantly to dsDNA rigidity, which is very misleading. It should be changed to point out the novelty of reference 4, which is to quantify the relative contributions of different factors to dsDNA stiffness. 

77: -> sequence-dependent. Also in this paragraph, the effect of base pair mismatches or bubbles on the persistence length can be discussed. It is more logical to follow this paragraph with the effects of chemically modified bases (96). 

81: Figure 1(c) is extremely hard to read.

85: The salt dependence is closely related to the second point. Screening of backbone charges by ions lowers the energetic cost of bending, thus the shorter persistence length.

90: No explanation is provided here. Temperature dependence shows that DNA bending is a thermally activated process over an energetic barrier. 

96: The authors dedicate a whole paragraph and a figure (g) to describe the single-molecule FRET experiment. I don’t think it is necessary considering that other experimental methods are not discussed to the same extent.    

106: In my opinion, the paragraph on experimental and computational methods for stiffness measurement should appear before discussing the results summarized in Figure 1. 

107: The authors are not doing justice to the vast literature by only citing one or two references for each methodology. For example, the ligation-based cyclization assay has been performed since 80s by Shore and Baldwin (1983) and many other groups (Vologodskii, Widom, etc.), and the FRET-based cyclization assays surfaced only recently, yet they only cite 21 for cyclization analysis. 

120: It is not clear what the word DNA spring refers to. Does it refer to a single strand (e.g., B)? 

122: This sentence compares the stiffness of A/B duplex to that of the stem, which is strange. The authors may want to show an intermediate state like Figure 3(b)B where the stem is subject to an unzipping force due to the bent dsDNA. I also think this bent dsDNA should be termed the DNA spring.

130: Some of these applications seem very promising. I think it is worth mentioning how easy or difficult it is to conjugate DNA strands to the biomolecule of interest. Readers would appreciate some discussion of the practical aspects.   

165: Figure 2(d) and (e). It is not clear how bending stiffness of DNA plays a role in these examples. As drawn, both examples contain long single strands that should provide enough slack to relieve any bending stress. 

202: The title is strange. Rigidity-based DNA sensors?

259: more -> move

269: It is not clear that hybridization of strand E will lead to an extended state. What mechanism causes linker duplexes to be parallel with each other? 

293: In Figure 5(a), shouldn’t the blue strand be discontinuous in multimer R? A nick in each duplex? 

311: bad wording. protein-DNA complexes as temperature and osmolarity sensors? Does osmolarity mean anything more than concentration?

329: It is stated that the persistence length increases with ionic strength, but Figure 6(b) shows the opposite.

329: decrease -> decreases

332: 3.4 is scientifically interesting, but I am extremely skeptical about the use of H-NS/DNA complex as an in vitro sensor of anything. Under what circumstances, would one choose to use this over a thermocouple in vitro? Please elaborate.  

345: bad wording. Just say “DNA springs as protein sensors”.

401: Is there a reason why the statement for the J factor is directly quoted? It seems a little odd given that there is nothing unique about that statement. This entire sentence can be removed.

405: circular DNA is not featured in Figure 8.

Reviewer 2 Report

Review of sensors-1007288

Mechanical flexibility of DNA…

The authors of this interesting and comprehensive review manuscript are interested in providing examples of nanotechnology and chemical biology applications of DNA structures and properties that exploit mechanical properties such as rigidity for practical purposes. Strengths of the manuscript include the number of references and broad base of interesting examples, including many helpful figures adapted or reproduced from the primary literature. This survey will likely be of interest to the field. The work is generally well-written and balanced.

The authors should consider the following minor points.

  1. While very well-referenced, two additional citations would seem appropriate in the introduction. These are the review of related concepts by Peters et al., (doi: 10.1017/S0033583510000077) and the discussion of controversy about DNA stiffness in the review by Vologodskii et al., (10.1093/nar/gkt396).
  2. A few typographical errors should be corrected:

Line 27: “put forth”

Line 77: “sequence-dependent”

Line 79: “poly(A)”
